# Immune transcriptomic changes in Australian Gulf War veterans

Natalie Eaton-Fitch[1,2]*, Etianne Martini Sasso[1,2], Sonya Marshall-Gradisnik[1,2]

1 National Centre for Neuroimmunology and Emerging Diseases, Health Group, Griffith University, Gold Coast, Australia, 2 Consortium Health International for Myalgic Encephalomyelitis, Griffith University, Gold Coast, Australia

* ncned@griffith.edu.au

## Abstract

### Background

Gulf War Illness (GWI) is a chronic multisystemic illness found in one-third of Gulf War Veterans. The aetiology of GWI is elusive; however, is strongly associated with exposure to multiple toxic agents, environmental exposures, and prophylactic medications or vaccinations. In the literature, disruption of the immune system and the presence of inflammation have been reported in GWI. In this novel study, we report gene expression-based analysis of a panel of 785 immune function related gene markers in GWI.

### Method

Ribonucleic acid (RNA) was extracted from peripheral blood mononuclear cells (PBMCs) isolated from n = 20 Australian GWI (CDC Case Definition and Kansas Criteria, 54.4 ± 0.74 years), and n = 15 healthy control (HC, 47.47 ± 2.91 years) participants. All participants were sex-matched (100% male). RNA gene expression was quantified using the NanoString® nCounter Immune Exhaustion panel and analysed using Rosalind Bio and IPA.

### Results

Thirty-three differentially expressed genes were identified, of which 21 were upregulated and 12 were downregulated in the GWI cohort. Upregulated genes included *SIGLEC1, BPI, MMP9, RSAD2, IFIT1/2, CEACAM1/3* and were associated with metabolic and cellular stress responses, while downregulated genes were associated with T cell receptor regions and humoral immune responses. Downregulated genes included *TRDV3, IGHG1, TRGV4, TRDV1/4,* and *IL7*. Gene set analysis revealed associations between gene expression and type I interferon signalling, natural killer receptors, T cell receptors, and tumour necrosis factor signalling. Pathway analysis

**Data availability statement:** All relevant data are within the manuscript and its Supporting Information files. The NanoString RNA-seq data are available at National Centre for Biotechnology Information Gene Expression Omnibus database using accession number GSE305935.

**Funding:** Griffith University Disability and Rehabilitation Program.

**Competing interests:** The authors have declared that no competing interests exist.

revealed the role of differentially expressed genes in neutrophil signalling and degranulation, toll-like receptor cascades, and the role of lipids/lipid rafts in infection.

## Conclusion

This investigation elucidates the potential role of immune dysregulation underlying GWI, emphasising the importance of immune exhaustion pathways in disease progression. Further investigations in a larger cohort may further elucidate or confirm these identified markers for potential screening or therapeutic applications in GWI.

## Introduction

Between August 1990 and February 1991, a coalition of 41 countries and approximately a million veterans participated in the Gulf War (GW), including 1,800 Australians. Following the first deployment, several epidemiological studies reported complex multi-systemic symptoms in veterans of the GW. The presence of chronic symptoms including fatigue, sleep disturbances, respiratory and epithelial complaints, neurocognitive disturbances, and body pain is referred to as GW Syndrome/Illness (GWI), a phenomenon categorised under the umbrella term chronic multisymptom illness [1]. GWI affects an estimated 25–32% of GW veterans (GWV) deployed to the GW in 1990–1991 [2]. Epidemiological reports in Australian GWV are consistent with research conducted in the United States (U.S.), United Kingdom (U.K.), Canadian, and French veterans reporting [3].

The pathomechanism of GWI remains elusive, as a consequence, there is no diagnostic test nor evidenced-based treatment available. Furthermore, the aetiology of GWI is not completely understood. Current evidence supports the hypothesis that a combination of toxic chemical and environmental agents, including insecticides, smoke from oil-well fires, pyridostigmine bromide (PB) result in a veteran developing GWI [4]. These toxic environmental and chemical agents are found to be statistically associated with immune system dysfunction [5,6]. While intracellular mechanisms remain diverse, agents including organophosphates and carbamate insecticides and PB, used prophylactically by veterans, as well as products from oil well fires, including particulate matter, heavy metals and polycyclic aromatic hydrocarbons are linked with chronic inflammation, oxidative stress and neuronal damage [7–10]. Therefore, demonstrating that no single exposure results in the occurrence of GWI.

One mechanism may include the inhibition of acetylcholinesterase by insecticides and PB resulting in an accumulation of acetylcholine which consequentially influences cellular metabolism and immunological functions, including reactive oxygen species (ROS), chronic inflammation, impaired cytotoxic pathways, cytokine production and immune cell activation [7–10]. As an example, an investigation into the effect of the pesticide permethrin with PB on a GWI mice model reported increased activation of both peripheral and brain adaptive immune responses [11]. Further, the accumulation of acetylcholine is further linked with oxidative stress. In an experimental GWI mice model, impaired cellular metabolism promoting ROS have been reported in

association with immune dysregulation [12]. This is further evidenced in diverse immunological studies reporting lymphocyte disturbances, altered lymphocyte subsets, interleukin (IL) and cytokine production, and production of antibodies are modified in veterans with GWI [13–16].

As many veterans continue to be affected in the decades following their return from the GW, research into the multisymptomatic pathomechanism of GWI is critical to avoid declining health in an aging population. Given the impact of immune responses linked to GWI symptomology, genetic variability that predisposes persistent inflammatory/immune alterations have been investigated [17]. Investigations into immune abnormalities provide valuable insights that may inform future biomedical research resulting in potential diagnostic tools and therapeutic interventions. This investigation aimed to elucidate transcriptome changes associated with immune exhaustion in Australian veterans with GWI.

## Method

### Participants

Australian GWV with multisystem symptoms and diagnosed with GWI were recruited with the assistance of the Gulf War Illness Association of Australia. Veterans with GWI fulfilled the CDC Case Definition [18] and Kansas Criteria [19] for GWI. Healthy controls (HC) reported an absence of disease and/or chronic diagnoses. Participants were screened according to their medical history and symptom presentation and were asked to report on quality of life measurements, including the 36-item short form health survey (SF-36) and World Health Organization Disability Assessment Schedule (WHODAS). All participants were males, aged between 18 and 65 years and non-smokers. Participants were not included in this current study if they reported a history of alcohol abuse, cardiovascular disease, thyroid disease, malignancies, insomnia, and another condition that may account for their symptoms. This investigation was approved by the Griffith University Human Research Ethics Committee (GU/2022/666). Research involving human research participants was performed per the Declaration of Helsinki and written consent was provided by all individuals prior to participation.

### Sample collection and preparation

Between 20–40 ml of whole blood was collected from each participant into ethylenediaminetetraacetic acid (EDTA) tubes via venepuncture at collection locations across Southeast Queensland and Northeast New South Wales from April 2022 to April 2024. All samples were collected between the hours of 7:00AM and 11:00AM from non-fasted participants. Four ml of EDTA whole blood was used for full blood count analysis.

Anonymised samples were delivered to the National Centre for Neuroimmunology and Emerging Diseases laboratory within four hours of collection. Peripheral blood mononuclear cells (PBMC) were isolated from whole blood by density gradient centrifugation using Ficoll (GE Healthcare, Uppsala, Sweden) as previously described [20]. PBMCs were stained with trypan blue (Invitrogen, Carlsbad, CA, USA) to determine cell count and viability. PBMCs were resuspended in fetal bovine serum (FBS) (Invitrogen Life Technologies, Carlsbad, CA, USA) containing 10% dimethyl sulfoxide (DMSO) and stored at -80°C until ribonucleic acid (RNA) extraction as previously described [21].

Frozen PBMCs were thawed and immediately pelleted by centrifugation in September 2024. Total RNA was isolated from PBMC pellets (5–10 x $10^6$ cells) using either the Trizol method (n = 2 samples) or a RNeasy Mini kit (Qiagen, Hilden, Germany) according to manufacturer instructions. The concentration and quality of the RNA were measured using NanoDrop ND-1000 spectrophotometer (Thermo Scientific, Massachusetts, US). RNA purity values were recorded with the mean and standard deviation being 1.99±0.10 for 260/280 and 1.33±0.43 for 260/230. While contaminants appeared present according to 260/230 values, all samples passed RNA binding density. The quality of the samples was confirmed using LabChip RNA Standard Sense Assay. All RNA samples returned a quality score above 8.9.

**RNA expression and NanoString®**

RNA expression analysis was determined using the NanoString® nCounter Immune Exhaustion gene expression panel according to manufacturer's instructions (NanoString Technologies, Seattle, WA, USA). A full list of investigated genes and their functional themes can be found in S1 Table. Batches of 12 separate samples were prepared according to the manufacturer's instructions.

Raw gene expression data was normalised against positive and negative controls. Normalisation and analyses were performed using Rosalind Bio (San Diego, CA, USA) and the following housekeeping genes *ABCF1, ALAS1, EEF1G, G6PD, GAPDH, GUSB, HPRT1, OAZ1, POLR1B, POLR2A, PPIA, RPL19, SDHA, TBP,* and *TUBB* (S1 Table). All raw gene expression data, normalised counts and quality check outputs cancan be found in S1 Table. Differential expression (fold change (FC) >1.5 or <−1.5 and a P-value <0.05) is reported between GWI with HC. Ingenuity Pathway Analysis (IPA) (Qiagen Digital Insights, California, USA) was used to interpret differential RNA expression in biological pathways and networks using algorithms developed for use by [22].

**Analysis**

The normality of participant data was determined using the Shapiro-Wilk test. Normally distributed continuous data was compared using the independent student's T test and non-normally distributed continuous data was compared using the Mann-Whitney U test. Age, body mass index (BMI), full blood count analysis, and quality of life measures are presented as mean ± standard error of mean (SEM) unless otherwise stated. Remaining participant demographics, including highest level of education and employment status were compared using the Chi-Square test and the Fisher's exact test. The effect of age and BMI on gene expression between cohorts was investigated using a general linear multivariate regression model. Participant data was analysed using SPSS (version 27) and GraphPad Prism (version 10). Significance is set at p < 0.05. Adjusted (Adj.) p-values are provided unless otherwise stated.

## Results

### Participants

This current study included n = 15 HC and n = 20 GWI. Participants with GWI were significantly older and reported higher BMI compared with HC (p = 0.013). There were no significant differences between participant cohorts in terms of the highest level of education achieved. There was a significant difference in the status of employment between cohorts (p = 0.027). Full blood count was determined for all participants and no significant differences were found between cohorts. All participants with GWI reported significantly lower SF-36 scores across all domains and higher scores of disability in all domains compared with HC. All clinical characteristics and demographics of participants are summarised in Tables 1 and 2.

The prevalence of symptoms reported by GWI participants is reported in Table 3. The most commonly reported symptoms included cognitive disturbances, pain, sleep, and neurosensory disturbances (94.7% of cohort).

### Differential gene expression

Of the 785 genes that are included in the NanoString Immune Exhaustion Panel, 601 genes passed background thresholding and normalisation. Differential expression of genes was filtered according to log fold change parameters −1.5 to 1.5 and a p-value of 0.05, resulting in the selection of 33 genes in GWI (Table 4 & Fig 1). Of the 33 selected genes, 21 were upregulated, and 12 were downregulated. Downregulated genes included *TRDV3 (T cell receptor delta variable 3)*, *IGHG1 (immunoglobulin (IG) heavy constant gamma 1)* and *TRDV4* (log$_2$FC = −1.664, p = 0.004; log$_2$FC = −1.438, p = 0.035; and log$_2$FC = −1.20, p = 0.04, respectively). Of the upregulated genes *SIGLEC1 (sialic acid binding Id-like lectin)* and *BPI (bactericidal/permeability increasing protein)* had the highest degree of change with log$_2$FC = 1.96 (p = 0.002), and log$_2$FC = 1.65 (p = 0.004), respectively. The full dataset outputs can be found in S2 Table.

**Table 1. Participant demographics.**

| | HC | GWI | P-value |
|---|---|---|---|
| | N(%) | N(%) | |
| Education n (%) | | | 0.110 |
| Primary School | 0 (0.0) | 0 (0.0) | |
| High School | 3 (20.0) | 7 (35.0) | |
| Undergraduate | 2 (13.3) | 8 (40.0) | |
| Postgraduate | 7 (46.7) | 3 (15.0) | |
| Other | 3 (20.0) | 2 (10.0) | |
| Employment n (%) | | | **0.027** |
| Full Time | 11 (73.3) | 9 (45.0) | |
| Part Time | 2 (13.3) | 0 (0.0) | |
| Casual | 1 (6.7) | 1 (5.0) | |
| Unemployed (other) | 1 (6.7) | 9 (45.0) | |
| Unemployed illness | 0 (0.0) | 1 (5.0) | |
| Sex n (%) | | | |
| Male | 15 (100%) | 20 (100.0%) | |

Categorical variables compared using chi-square test. Data presented as n (%). *Abbreviations: HC, healthy control; GWI, Gulf War Illness.*

## Gene set analysis

The change in regulation within each gene set relative to the baseline was described using gene set analysis (GSA), both undirected enrichment score (UES) and directed enrichment score (DES) for the top 10 gene sets are presented in Table 5. GSA was obtained from Rosalind Bio. Differentially expressed genes in GWI were associated with Type I Interferon (UES = 1.7835, DES = 1.7386), NK receptors (UES = 1.6857, DES = −0.1878), fatty acid metabolism (UES = 1.4838, DES = −1.0185), tumour necrosis factor (TNF) signalling (UES = 1.3493, DES = 0.9285), and T cell receptor (UES = 1.319, DES = −0.7643). The full dataset outputs can be found in S2 Table.

## Cell type abundance

The abundance of cell populations was calculated according to the expression of cell marker genes using Rosalind Bio. Hierarchical cluster analysis observations demonstrate similar distributions within cohorts for major immune cell populations (Fig 2). While not significant, veterans with GWI had a slightly higher mean abundance of T cells marker genes compared with HC and was accompanied by lower Treg cells. Neutrophils marker genes were also found to be lower in GWI relative to HC; however, this was not significant. Abundance scores for cell types are shown in S3 Table.

## Pathways and disease functions

A summary of IPA analysis, including the top five biological functions and canonical pathways are presented in Table 6. The top five biological functions in GWI include childhood-onset systemic lupus erythematosus (SLE), ATAD3A-related type I interferonpathy, SLE, activation of leukocytes, and antineutrophil cytoplasmic antibody associated vasculitis (all p < 0.0001). *RSAD2* and *SIGLEC1* were consistently reported across all biological functions. The top five canonical pathways in GWI include neutrophil extracellular trap signalling pathway (p = 0.0023), neutrophil degranulation (p = 0.0031), role of lipids/lipid rafts in influenza (p = 0.0045), chaperone mediated autophagy signalling pathway (p = 0.0055) and toll-like receptor cascades (p = 0.006). The complete pathways and disease functions output can be found in S4 Table.

**Table 2. Participant demographics, full blood analysis and quality of life.**

| | HC | | | GWI | | | P-value |
|---|---|---|---|---|---|---|---|
| | Mean | SEM | SD | Mean | SEM | SD | |
| Age | 47.47 | 2.91 | 11.27 | 54.4 | 0.74 | 3.29 | **0.013** |
| BMI | 25.96 | 1.09 | 4.23 | 31.44 | 1.62 | 7.23 | **0.013** |
| Full blood count analysis | | | | | | | |
| WCC (x10⁹/L) | 6.11 | 0.31 | 1.19 | 6.78 | 0.33 | 1.47 | 0.158 |
| Lymphocyte (x10⁹/L) [a] | 1.88 | 0.19 | 0.77 | 2.03 | 0.14 | 0.63 | 0.214 |
| Neutrophils (x10⁹/L) | 3.49 | 0.20 | 0.79 | 3.98 | 0.26 | 1.15 | 0.163 |
| Monocytes (x10⁹/L) | 0.51 | 0.05 | 0.18 | 0.56 | 0.03 | 0.15 | 0.287 |
| Eosinophils (x10⁹/L) [a] | 0.19 | 0.04 | 0.14 | 0.16 | 0.02 | 0.09 | 0.856 |
| Basophils (x10⁹/L) | 0.05 | 0.01 | 0.02 | 0.05 | 0.004 | 0.02 | 0.924 |
| Platelets (x10⁹/L) | 260.13 | 10.97 | 42.49 | 263.55 | 14.91 | 66.66 | 0.863 |
| RCC (x10¹²/L) | 5.08 | 0.08 | 0.32 | 5.07 | 0.09 | 0.39 | 0.932 |
| Haematocrit | 0.44 | 0.01 | 0.02 | 0.45 | 0.01 | 0.03 | 0.082 |
| Haemoglobin (g/L) | 149.07 | 2.24 | 8.68 | 151.10 | 2.30 | 10.31 | 0.542 |
| SF-36 | | | | | | | |
| General Health | 71.38 | 4.60 | 17.81 | 37.25 | 4.12 | 18.44 | **<0.001** |
| Physical Functioning[a] | 96.0 | 2.02 | 7.83 | 57.50 | 5.99 | 26.83 | **<0.001** |
| Role Physical | 82.92 | 9.46 | 36.63 | 45.31 | 6.44 | 28.81 | **0.002** |
| Role Emotional | 95.56 | 2.68 | 10.38 | 49.99 | 7.13 | 31.88 | **<0.001** |
| Pain[a] | 90.67 | 3.40 | 13.17 | 38.0 | 4.54 | 20.32 | **<0.001** |
| Mental Health | 83.67 | 3.57 | 13.82 | 47.50 | 5.37 | 24.03 | **<0.001** |
| Vitality | 75.83 | 3.39 | 13.12 | 40.63 | 3.06 | 13.68 | **<0.001** |
| Social Functioning | 97.50 | 1.81 | 7.01 | 43.75 | 6.87 | 30.75 | **<0.001** |
| WHO DAS | | | | | | | |
| Understanding & communication | 8.06 | 3.56 | 13.78 | 39.79 | 5.17 | 23.12 | **<0.001** |
| Mobility | 2.33 | 1.61 | 6.23 | 33.0 | 5.39 | 24.08 | **<0.001** |
| Self-care[a] | 0.83 | 0.83 | 3.23 | 15.0 | 3.37 | 15.09 | **<0.001** |
| Relationships | 5.42 | 2.58 | 9.98 | 45.0 | 5.82 | 26.02 | **<0.001** |
| Life activities | 2.08 | 1.45 | 5.62 | 36.56 | 5.29 | 23.67 | **<0.001** |
| Participation in society | 5.21 | 2.77 | 10.74 | 44.37 | 6.26 | 28.01 | **<0.001** |

Continuous variables compared using Mann Whitney U test or T test. [a] Denotes continuous variables compared using Mann Whitney U test. The WHO DAS domain for participation in work/school was omitted given the high number of participants reporting unemployment. Data presented as mean, SEM and SD. *Abbreviations: HC, healthy control; GWI, Gulf War Illness; BMI, body mass index; SD, standard deviation; SEM, standard error of mean; WCC, white cell count; RCC, red cell count; SF-36, 36 item short-form health survey; WHO, World Health Organization; DAS, disability assessment schedule.*

## Network analysis

Interaction network analysis was performed using IPA. This analysis demonstrates the interactions between molecules and the dataset imported. One network was exported, with a score of 16 (Fig 3). Analysis of this network, consisting of five focus molecules, was associated with cell-to-cell signalling and interactions, haematological system development and function, and immune cell trafficking. Focus molecules identified were *BPI* (p = 0.004), *MMP9* (p = 0.0004), *RSAD2* (p = 0.004), *SIGLEC1* (p = 0.002), and *TRDV3* (p = 0.005). Top upstream regulators were identified to be ATAD3A (p < 0.0001), TNF (family, p < 0.0001), RNY3 (p < 0.0001), SOD1 (p < 0.0001) and CSF1 (p < 0.0001). Molecules in causal

**Table 3. Symptom prevalence in Australian GWI participants.**

| Symptom | n (%) |
|---|---|
| Post-exertional malaise | 17 (89.5) |
| Cognitive disturbances | 18 (94.7) |
| Pain | 18 (94.7) |
| Sleep | 18 (94.7) |
| Neurosensory, perceptual, and motor disturbances | 18 (94.7) |
| Immune | 9 (47.4) |
| Respiratory | 9 (47.4) |
| Gastrointestinal | 15 (78.9) |
| Urinary disturbances | 11 (57.9) |
| Cardiovascular manifestations | 11 (57.9) |
| Thermoregulatory disturbances | 11 (57.9) |

Data presented as n (%) for those reporting experiencing the symptom. Missing data n = 1. *Abbreviations: GWI, Gulf War Illness; n, sample number.*

network identified ATAD3A (p < 0.0001), RNY3 (p < 0.0001), Jak (family) (p < 0.0001), and type I IFN genes (p < 0.0001). Network analysis outputs can be found in S4 Table.

## Discussion

This novel study investigates altered gene expression related to immune function in Australian veterans with GWI compared with HC. Briefly, 33 differentially expressed genes were identified, 21 of which were upregulated and 12 were downregulated. Of the differentially expressed genes, *TRDV3*, *IGHG1*, *TRGV4*, *TRDV1*, and *TRDV2* returned as the top five downregulated, while *SIGLEC1, BPI, MMP9, RSAD2,* and *CEACAM1* were the top five upregulated genes. To the authors knowledge this present research is the first to conduct an analysis on immune exhaustion and inflammation markers simultaneously using NanoString Technology in this cohort.

Previous studies have investigated genetic markers in GWI with the identification of nerve agent susceptible genes including *PON1* (paraoxonase-1) and *BChE* (butyrylcholinesterase) [23,24] as well as the neurodegeneration gene *APOE* (apolipoprotein) [25]. Meanwhile, immune profiling has identified altered expression of inflammatory markers including *IL-1β, TNFα, MMP-2, CCL12,* and *EGF*, some of which aligning with the present investigation [26]. A previous study employing logistic regression modelling created a prediction model of GWI risk associated with genetic variability in *TGF* (rs1800469, p = 0.009), *IL6R* (rs8192284, p = 0.004) and *TLR4* (rs4986791, p = 0.013) [17]. While variability within these mentioned genes was not identified in the present study, toll-like receptor cascades and IL signalling were significantly associated with the top differential genes identified in the present manuscript. Further research has also reported an associated with human leukocyte antigen (HLA) allele DRB1*13:02 [27]. Previous research and the present study identified potential markers worthy of further investigation to elucidate the role of immune disturbances in the pathomechanism of GWI.

TCR genes (*TRDV3*, *TRDV1*, *TRDV2*, *TRGV4* and *TRGC1*) were downregulated in Australian veterans with GWI compared with HC, suggesting potential consequences that result in altered downstream T cell activity and cytotoxic function. This is supported by the abovementioned GSA data whereby TCR signalling returned a negative DES, indicating a functional reduction. A downregulation of TCRs will impair T cell responsiveness to antigen stimulation through impaired T cell to target cell interactions and a decrease in downstream intracellular signalling cascades [28]. TCR downregulation accompanied by reduced IL-2 signalling, demonstrated through a negative DES above, further supports the suggestion of impaired T cell survival and proliferation in GWI [29].

**Table 4. Differential gene expression.**

| Gene | Description | log2FC | P-value |
|---|---|---|---|
| **Downregulated** | | | |
| TRDV3 | T cell receptor delta variable 3 | −1.6643 | 0.0049 |
| IGHG1 | Immunoglobulin heavy constant gamma 1 | −1.4380 | 0.0345 |
| TRGV4 | T cell receptor gamma variable 4 | −1.2019 | 0.0496 |
| TRDV1 | T cell receptor delta variable 1 | −1.1313 | 0.0018 |
| TRDV2 | T cell receptor delta variable 2 | −1.1312 | 0.0269 |
| IL7 | Interleukin 7 | −0.8934 | 0.0170 |
| IGHV4–59 | Immunoglobulin heavy variable 4–59 | −0.7757 | 0.0351 |
| EHHADH | 3-hydroxyacyl CoA dehydrogenase | −0.7612 | 0.0107 |
| IDO1 | Indoleamine 2,3-dioxygenase 1 | −0.7149 | 0.0473 |
| CXCR6 | chemokine (C-X-C motif) receptor 6 | −0.7101 | 0.0160 |
| TRGC1 | T cell receptor gamma constant 1 | −0.7017 | 0.0098 |
| SESN2 | Sestrin 2 | −0.6490 | 0.0221 |
| **Upregulated** | | | |
| SIGLEC1 | Sialic acid binding Ig-like lectin 1, sialoadhesion | 1.9591 | 0.0022 |
| BPI | Bactericidal/permeability-increasing protein | 1.6511 | 0.0045 |
| MMP9 | Matrix metallopeptidase 9 | 1.5475 | 0.0004 |
| RSAD2 | Radical S-adenosyl methionine domain containing 2 | 1.5471 | 0.0037 |
| CEACAM1 | Carcinoembryonic antigen-related cell adhesion molecule 1 (biliary glycoprotein) | 1.4122 | 0.0022 |
| IFIT1 | Interferon-induced protein with tetratricopeptide repeats 1 | 1.3933 | 0.0111 |
| IFIT3 | Interferon-induced protein with tetratricopeptide repeats 3 | 1.3519 | 0.0075 |
| CXCL1/2/3 | Chemokine (C-C motif) ligand 1/2/3 | 1.3366 | 0.0292 |
| PTGS2 | Prostaglandin-endoperoxide synthase 2 (prostaglandin G/H synthase and cyclooxygenase) | 1.3063 | 0.0287 |
| CEACAM3 | Carcinoembryonic antigen-related cell adhesion molecule 3 | 1.2773 | 0.0045 |
| ITGB3 | Integrin beta 3 (platelet glycoprotein IIIa antigen CD61) | 1.2059 | 0.0019 |
| EGF | Epidermal growth factor | 1.0986 | 0.0251 |
| LTBP1 | Latent transforming growth factor beta binding protein 1 | 1.0477 | 0.0074 |
| GREM2 | Gremlin 2 DAN family BMP antagonist | 0.9805 | 0.0438 |
| OAS3 | 2'-5'-oligoadenylate synthetase 3 (100kDa) | 0.9389 | 0.0129 |
| IL1B | Interleukin 1 beta | 0.9364 | 0.0483 |
| ELOVL7 | ELOVL fatty acid elongase 7 | 0.8905 | 0.0079 |
| MX1 | MX dynamin-like GTPase 1 | 0.8547 | 0.0191 |
| TNFAIP3 | Tumor necrosis factor alpha-induced protein 3 | 0.7483 | 0.0209 |
| FCAR | Fc fragment of IgA receptor | 0.6453 | 0.0153 |
| MX2 | MX dynamin-like GTPase 2 | 0.5934 | 0.0115 |

Data extracted from Rosalind Bio. Descriptions extracted from the National Institutes of Health (NIH) National Library of Medicine gene database. Abbreviations: FC, fold change; TRDV3, T cell receptor delta variable 3; IGHG1, Immunoglobulin heavy constant gamma 1; TRDV2, T cell receptor delta variable 2; TPSAB1/B2, tryptase alpha/beta 1; TRGV4, T cell receptor gamma variable 4; TRDV1, T cell receptor delta variable 1; IL7, Interleukin 7; IGHV4−59, immunoglobulin heavy variable 4−59; IDO1, indoleamine 2 3-dioxygenase 1; EHHDAH, enoyl-CoA hydratase/3-hydroxyacyl CoA dehydroge- nase; CXCR6 chemokine (C-X-C motif) receptor 6; TRGC1, T cell receptor gamma constant 1; SESN2, sestrin 2; SIGLEC, sialic acid binding Ig-like 1; BPI, bactericidal/permeability-increasing protein; RSAD2, radical S-adenosyl methionine domain containing 2; MMP9, Matrix metallopeptidase 9; IFIT1, interferon-induced protein with tetratricopeptide repeats 1; CEACAM1, Carcinoembryonic antigen-related cell adhesion molecule 1; IFIT3, interferon- induced protein with tetratricopeptide repeats 3; CXCL1/2/3, CXCL1, CXCL2 (MIP-2 alpha), Chemokine (C-C motif) ligand 1/2/3;CEACAM3, Carcinoem- bryonic antigen-related cell adhesion molecule 3; PTGS2, prostaglandin-endoperoxide synthase 2; ITGB3, integrin beta 3; LTBP1, latent transforming growth factor beta binding protein 1; EGF, epidermal growth factor; OAS3, 2-;5- oligoadenylate synthetase 3; ELOVL7, ELOVL fatty acid elongase 7; MX1, MC dynamin-like GTPase 1; TNFAIP3, Tumour necrosis factor, alpha-induced protein 3; GREM2, gremlin 2; IL1B, interleukin 1 beta; FCAR, Fc fragment of IgA receptor; MX2, MX dynamin-like GTPase 2.

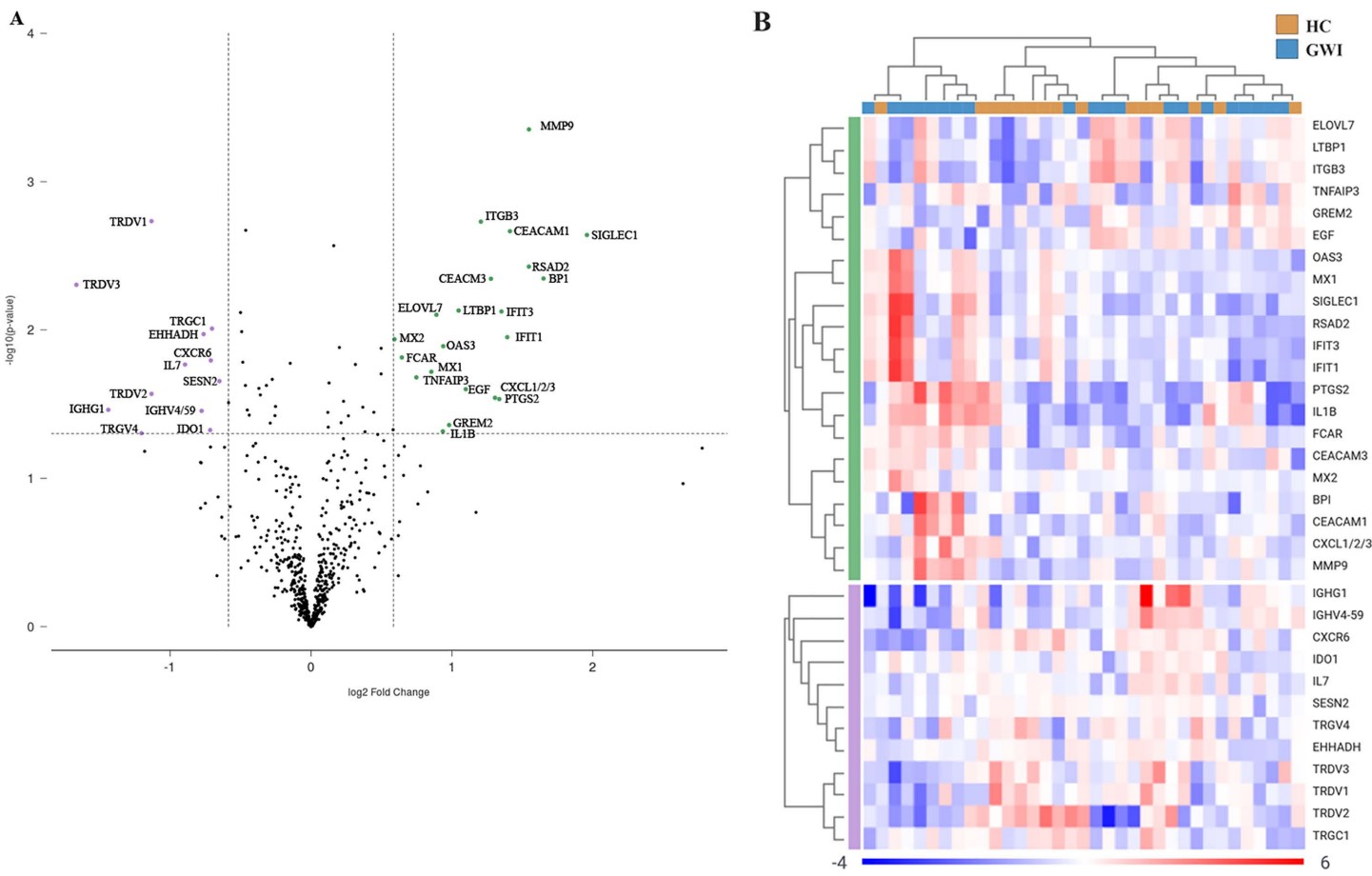

**Fig 1. Differentially expressed genes in Australian GWI.** (A) volcano plot displaying statistical significance (log10(p-value) on the y-axis, and log2 fold change on the x-axis. Selected genes meeting filter criteria are presented as those down-regulated (≤−1.5) and those upregulated (≥1.5). (B) heatmap of selected genes representing log2 normalised expression values from −4 to 6. Red indicates high levels of expression, while blue indicates low levels of expression. Clusters are organised according to upregulated or downregulated genes by participant cohort. Green indicates upregulated genes, while purple indicates downregulated genes. Figure exported from Rosalind Bio. *Abbreviations: HC, healthy control; GWI, Gulf War Illness; TRDV3, T cell receptor delta variable 3; IGHG1, Immunoglobulin heavy constant gamma 1; TRDV2, T cell receptor delta variable 2; TPSAB1/B2, tryptase alpha/beta 1; TRGV4, T cell receptor gamma variable 4; TRDV1, T cell receptor delta variable 1; IL7, Interleukin 7; IGHV4−59, immunoglobulin heavy variable 4−59; IDO1, indoleamine 2 3-dioxygenase 1; EHHDAH, enoyl-CoA hydratase/3-hydroxyacyl CoA dehydrogenase; CXCR6 chemokine (C-X-C motif) receptor 6; TRGC1, T cell receptor gamma constant 1; SESN2, sestrin 2; SIGLEC, sialic acid binding Ig-like 1; BPI, bactericidal/permeability-increasing protein; RSAD2, radical S-adenosyl methionine domain containing 2; MMP9, Matrix metallopeptidase 9; IFIT1, interferon-induced protein with tetratricopeptide repeats 1; CEACAM1, Carcinoembryonic antigen-related cell adhesion molecule 1; IFIT3, interferon-induced protein with tetratricopeptide repeats 3; CXCL1/2/3, CXCL1, CXCL2 (MIP-2 alpha), Chemokine (C-C motif) ligand 1/2/3;CEACAM3, Carcinoembryonic antigen-related cell adhesion molecule 3; PTGS2, prostaglandin-endoperoxide synthase 2; ITGB3, integrin beta 3; LTBP1, latent transforming growth factor beta binding protein 1; EGF, epidermal growth factor; OAS3, 2-;5- oligoadenylate synthetase 3; ELOVL7, ELOVL fatty acid elongase 7; MX1, MC dynamin-like GTPase 1; TNFAIP3, Tumour necrosis factor, alpha-induced protein 3; GREM2, gremlin 2; IL1B, interleukin 1 beta; FCAR, Fc fragment of IgA receptor; MX2, MX dynamin-like GTPase 2.*

Downregulation of TCRs and impaired T cell function are reported in various pathologies, including cancer, autoimmune diseases and infectious diseases [30]. Similarly, TCR dysregulation may contributed to the changes in T cell subsets and pro-inflammatory responses observed in GWI. Immune profiling and function have been investigated and have reported altered T- and NK lymphocyte profiles and impaired cytotoxic function, with emphasis on elevated T lymphocyte populations of veterans with GWI compared with controls [13,14,31]. Whether the downregulation of TCRs is correlated to a

**Table 5. Gene set analysis for genes differentially expression in Australian GWI.**

| GSA | UES | DES |
| --- | --- | --- |
| Type I Interferon | 1.7835 | 1.7386 |
| NK Receptors | 1.6857 | −0.1878 |
| Fatty Acid Metabolism | 1.4838 | −1.0185 |
| TNF Signalling | 1.3493 | 0.9285 |
| TCR Signalling | 1.319 | −0.7643 |
| Hypoxia Response | 1.2441 | 0.9354 |
| IL-7 Signalling | 1.2037 | −0.5323 |
| PPAR Signalling | 1.1884 | −1.1031 |
| Chemokine Signalling | 1.1768 | −0.8164 |
| Other IL Signalling | 1.1676 | 0.7214 |

Data extracted from Rosalind Bio. Abbreviations: GSA, gene set analysis; UES, undirected enrichment score; DES, directed enrichment score; NK, natural killer; TNF, tumour necrosis factor, TCR, T cell receptor; IL, interleukin; PPAR, peroxisome proliferator-activated receptors.

potential elevation in T cell populations are unknown. This current investigation did not directly quantify lymphocyte cell numbers and cell profiling according to differentially expressed genes did not significantly differ between GWI and HC cohorts. Instead, future investigations may aim to concurrently analyse gene expression with phenotyping. Notably, treatment with PB, a medication routinely administered as a prophylactic during the Gulf War, can impede T cells through the cholinergic anti-inflammatory pathway (CAP) resulting in the suppression of T cell activity [32]. However, the implications of long-term or excessive PB, as reported by GWV, on T cell activity is unknown. Nevertheless, reduced TCR functioning in GWI is a novel finding and supports the need for further investigations into immune disturbances in Australian GWV.

Upregulated interferon-related genes including *RSAD2*, *IFIT1*, *IFIT3*, *MX1*, *MX2* and *OAS3* suggests chronic activation of antiviral and inflammatory pathways. This is further supported by GSA data whereby type I interferon associated genes returned a positive DES, as seen above. Type I interferon activation suggests persistent immune activation. While elevated interferon signalling may also suggest automimicry in this cohort of GWI, the downregulation of *IGHG1* and *IGHV4–59* suggests reduced humoral immune responses and impaired antibody production, not reminiscent of the presence of autoimmunity which is inconsistent in GWI research [33,34]. Upregulation of other inflammatory and immune activation markers reported in the present manuscript, such as *IL1B*, *PTGS2*, *TNFAIP3*, *CXCL1/2/3* and *FCAR* aligns further with chronic inflammation or innate immune activation [35–38]. In further support of our research, a previous investigation reported elevated levels of IFN-y- in addition to IL-2-producing CD4 + cells and elevated *in vitro* levels of IL-10-producing CD4 cells compared with non-symptomatic GWV [39]. While plasma levels of IL-6 and C-reactive protein (CRP) are also found to be increased in veterans with GWI [40]. Therefore, the present research, in conjunction with the existing literature, supports the role of chronic inflammation in the pathomechanism of GWI.

The downstream implications of the preceding exposures and resulting chronic inflammation are potentially associated with metabolic disturbances reported in GWI. Results of the present investigation suggest metabolic dysfunction evidenced through differential expression of genes including *EGF*, *ITGB3*, *LTBP1*, *GREM2, SESN2*, and *ELOVL7*. For example, the downregulation of *SESN2,* encoding for sestrin 2, suggests mitochondrial deoxyribonucleic acid damage, oxidative stress, and hypoxia reported in human disease, including those that are neurodegenerative [41]. Previous investigations have reported that impaired mitochondrial function is associated with symptom severity in veterans with GWI [42]. A review of research on inflammation and ROS suggests that inflammatory mediators, such as those reported in GWI cohorts, may potentially exacerbate metabolic dysfunction and fatty acid oxidation [43]. Suggesting that inflammation and mitochondrial dynamics are interconnected in disease. Additionally, Bryant *et al.* reported that altered cellular metabolism

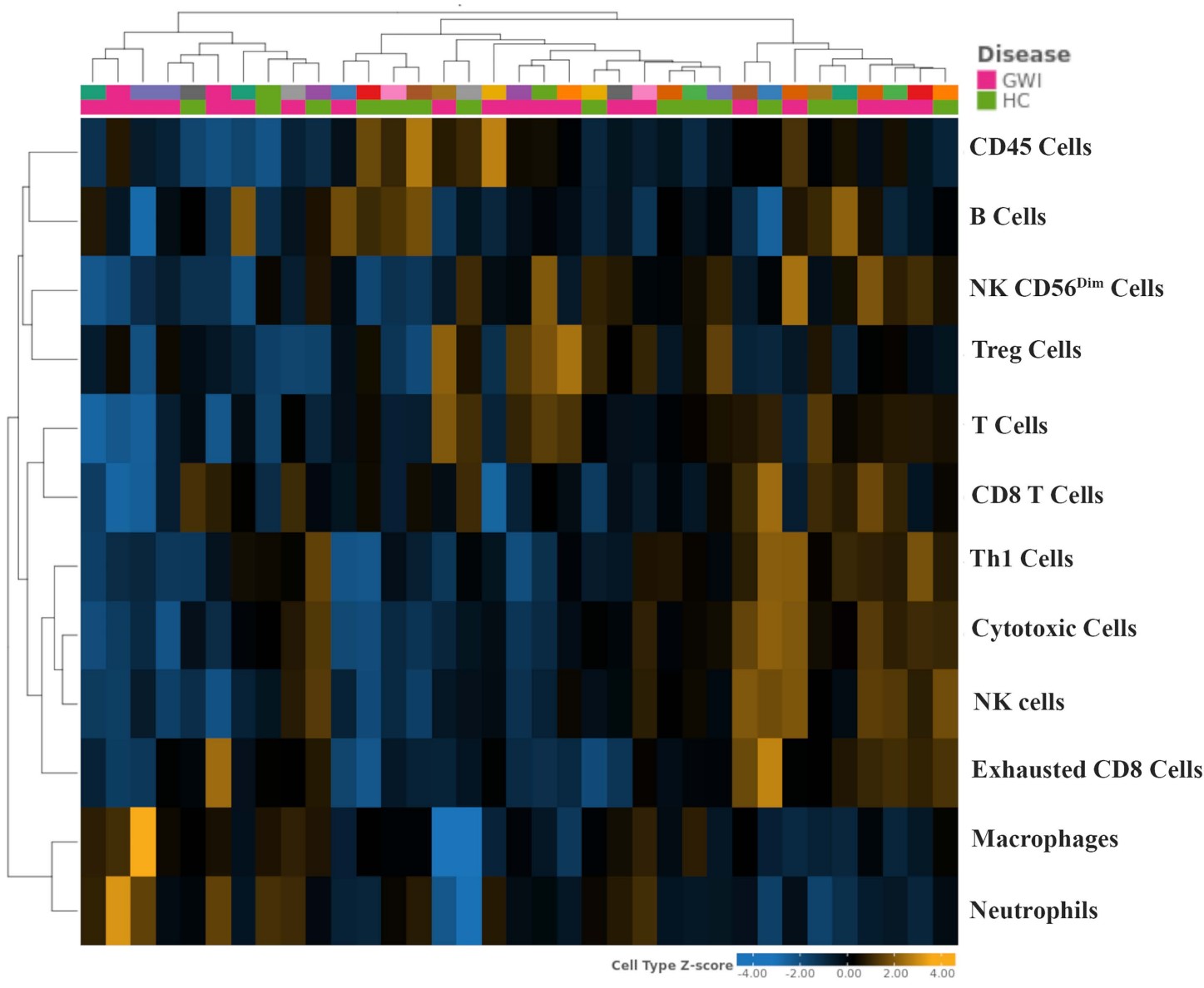

**Fig 2. Cell profiles and gene expression.** (A) Heatmap extracted from Rosalind Bio. Cell type z-score for cell populations were populated for samples collected from GWI and HC. *Abbreviations: NK, natural killer; Treg, T regulatory; Th, T helper.*

in a GWI mouse model may promote inflammatory processes in veterans; however, further protein- and functional-level research is required to determine whether this is supported by the present study. The downstream effects of genes including *SESN2* and others listed above on cellular metabolism may indeed exacerbate inflammation aligning with Bryant *et al* [26]. Other research has reported on genetic variants linked with mitochondrial disturbances, such as *BChE* [24]. While transcriptomics research in GWI mice have also reported differential expression of genes important for mitochondrial respiration, oxidative phosphorylation and electron transport chain [44], the top differentially expressed genes are not replicated by the present investigation. Differences in the literature may be linked to the use of a GWI mouse model [26,44] or samples provide by veterans with GWI [42,45], therefore, posing a barrier when interpreting data between models.

**Table 6.  Top biological functions and pathways in Australian GWI.**

| Functions | P-value | Molecules |
|---|---|---|
| Childhood-onset SLE | <0.0001 | RSAD2,SIGLEC1 |
| ATAD3A-related type I interferonopathy | <0.0001 | RSAD2,SIGLEC1 |
| SLE | <0.0001 | RSAD2,BPI,MMP9,SIGLEC1 |
| Activation of leukocytes | <0.0001 | RSAD2,BPI,MMP9,SIGLEC1 |
| Antineutrophil cytoplasmic antibody-associated vasculitis | <0.0001 | RSAD2,MMP9,SIGLEC1 |
| **Pathways** | **P-value** | **Ratio** |
| Neutrophil Extracellular Trap Signalling Pathway | 0.0023 | 0.00489 |
| Neutrophil degranulation | 0.0031 | 0.00419 |
| Role of Lipids/Lipid Rafts in the pathogenesis of influenza | 0.0045 | 0.0417 |
| Chaperone Mediate Autophagy Signalling Pathway | 0.0055 | 0.00314 |
| Toll-like Receptor Cascades | 0.006 | 0.0312 |

Data extracted from IPA. Ratio is calculated as the number of molecules in a given pathway that meets cutoff criteria, divided by the total number of molecules that make up that pathway and that are in the reference set. *Abbreviations: SLE, systemic lupus erythematosus; SIGLEC, sialic acid binding Ig-like 1; BPI, bactericidal/permeability-increasing protein; RSAD2, radical S-adenosyl methionine domain containing 2; MMP9, Matrix metallopeptidase 9.*

Overall, this present research is supported by the literature reporting a decline in metabolic function in veterans with GWI and therefore provides avenues for future research [45–47].

Further, GSA suggested the potential occurrence of impaired fatty acid metabolism in the present GWI cohort. While the BMI of veterans with GWI was significantly higher compared with HC, previous research using a GWI mouse model demonstrated mitochondrial lipid changes in the brains and plasma [48]. This mouse model was exposed to GW agents PB and permethrin, therefore, it can be hypothesised that mitochondrial disturbances reported in this present manuscript are potentially linked to exposures of GWV and the role of fatty acid metabolism disturbances cannot be limited to BMI in this cohort. This is further supported by multivariate analysis undertaken to determine association between BMI and gene expression which found potential associated with the expression of *IGHV4/59* and *CEACAM3* and not genes related to fatty acid metabolism (S2 Table). Nevertheless, further research is warranted to determine the impact of high BMI on the regulation of metabolic genes, as the sample size of this research did not allow data stratification on BMI.

Currently, there is no validated biomarker for GWI for diagnosis or to determine risk susceptibility. The identification of biological markers could help to refine illness definition, better detect, predict or distinguish subgroups of GWI, and ultimately lead to the development of hypothesis-driven and evidence-based treatments to improve health outcomes of veterans. While the occurrence of immune exhaustion was investigated using a targeted genomic panel, typical immune exhaustion markers such as Programmed Death-1 (PD-1), lymphocyte activating gene (LAG), T cell immunoreceptor ITIM domain 3 (TIM-3) and cytotoxic T lymphocyte associated proteins (CTLAs) [49], were not identified as differentially expression. Rather, the results of the current investigation suggest the role of chronic immune activation and inflammation in Australians with GWI. Given the impact of immune responses linked to GWI symptomology, genetic variability that causes persistent inflammatory or immune alterations may be essential for further research into diagnostic tools or targeted pharmacotherapeutic intervention. This present research provides the foundations to facilitate further analysis for potential biomarker identification specific to GWI. This current manuscript further elucidates the role of immunological disturbances in the pathomechanism of GWI. Importantly, the mechanisms involved in the immune disturbances reported in GWI are potentially linked with immunotoxicity caused by exposures during the Gulf War resulting in lymphocyte dysfunction, increased oxidative stress, and dysregulation of immune signalling pathways [5,6,50].

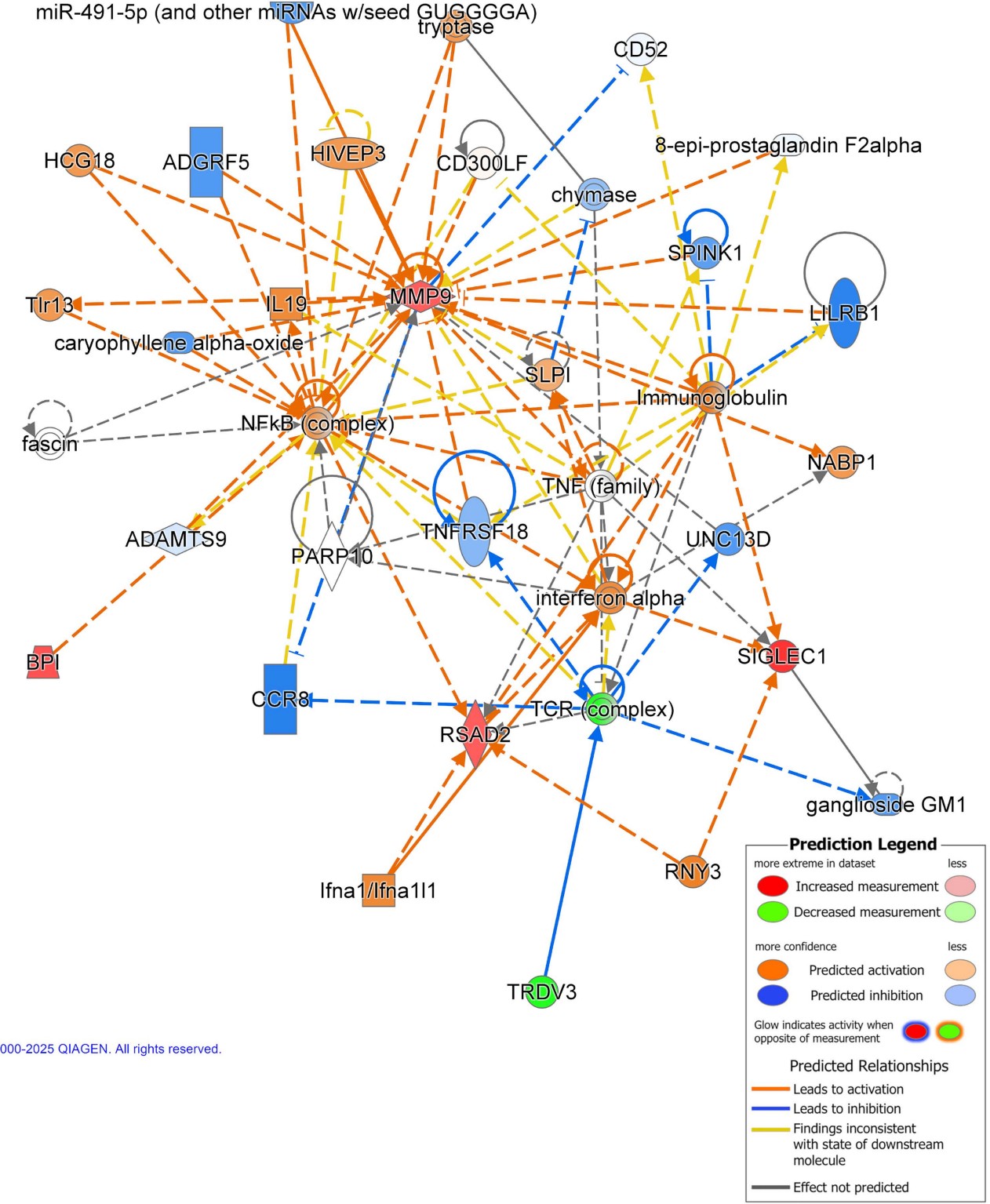

**Fig 3. Network analysis in Australian GWI.** Gene interaction network map consisting of top filtered differentially expressed genes. Genes are organised according to subcellular space. Network analysis score = 16. *Abbreviations: TRDV3, T Cell Receptor Delta Variable 3; RNY3, RNA, Ro-Associated Y3; SIGLEC1, Sialic Acid Binding Ig Like Lectin 1; NABP1, Nucleic Acid Binding Protein 1; RSAD2, Radical S-Adenosyl Methionine Domain Containing*

*2; CCR8, C-C Motif Chemokine Receptor 8; TCR, T Cell Receptor; BPI, Bactericidal/Permeability-Increasing Protein; ADAMTS9, ADAM Metallopeptidase With Thrombospondin Type 1 Motif 9; PARP10, Poly(ADP-Ribose) Polymerase Family Member 10; TNFRSF18, Tumor Necrosis Factor Receptor Superfamily Member 18; TNF, Tumor Necrosis Factor; SLPI, Secretory Leukocyte Peptidase Inhibitor; NFkB, Nuclear Factor Kappa-Light-Chain-Enhancer of Activated B Cells; IL19, Interleukin 19; ADGRF5, Adhesion G Protein-Coupled Receptor F5; HCG18, HLA Complex Group 18; TIR13, Likely refers to TIR domain-containing protein 13, though this is not a well-characterized molecule; MMP9, Matrix Metallopeptidase 9; SPINK1, Serine Peptidase Inhibitor, Kazal Type 1; LILRB1,Leukocyte Immunoglobulin-Like Receptor Subfamily B Member 1; HIVEP3, Human Immunodeficiency Virus Type I Enhancer Binding Protein 3. Figure constructed using IPA, Qiagen.*

This current investigation is not without limitations. The small cohort sizes limit stratification of cohorts according to clinical presentation, age, BMI, and other potential confounding factors. This emphasises the need for further investigations with larger cohorts to differentiate potential immune subtypes and identify biomarkers for stratification. Nevertheless, effect sizes calculated for each differentially expressed gene were found to be moderate to large demonstrating a sufficient sample size to support these findings (S2 Table). With further investigations incorporating larger sample sizes and protein-level validation may also be considered for future research. Given the significant differences in age between the cohorts, a multivariate analysis was performed to determine any potential effect on gene expression. We report that the effect of age on gene expression was non-significant, excluding one gene being *IGHV4/59* (S2 Table). It is important to highlight that while this current investigation raised disease pathways associated with SLE, no participants reported a diagnosis of an autoimmune condition. This current investigation serves as the basis to justify further larger investigations to identify immunological biomarkers in GWI. Further, the Immune Exhaustion panel developed by NanoString biases expression analysis to a small selection of genes. While this technology provides sensitive data, future analysis may consider the validation of gene expression analysis using untargeted RNA expression analysis with quantitative polymerase chain reaction experiments to confirm findings.

## Conclusion

This investigation reports immune transcriptome changes in Australian veterans with GWI using NanoString Immune Exhaustion panel. Altered gene expression identified in this study indicates changes to both innate and adaptive immune responses with evidence of metabolic stress, and IL signalling disturbances. The findings of this present research suggest chronic inflammation is a potential mechanism underpinning symptom presentation of GWI. Markers of immune exhaustion were not statistically different in Australian GWI participants. Moreover, these findings contribute to the growing body of literature on the pathomechanism of immunological disturbances in GWI and may facilitate further research in the identification of diagnostic or therapeutic targets.

## Supporting information

**S1 Table. This file contains pre-processed gene expression data collected prior to statistical or pathway analyses.** Information and data was generated using NanoString nCounter Immune Exhaustion Panel and Rosalind Bio. Inside the file contains a full list of genes and probes included in the NanoString nCounter Immune Exhaustion Panel including probe IDs, gene symbols and probe sequences (T1); lists corresponding functional themes of genes incorporated within the panel supporting predetermined information used for gene set and pathway analysis (T2); a summary of panel coverage according to genes within the panel contributing to immune-related pathways or cell types (T3); raw gene expression for housekeeping genes (T4); normalised RNA expression counts subsequent to positive control normalisation and background removal for all genes fulfilling thresholding requirements (T5); raw RNA gene expression counts prior to normalisation or quality control (T6) and; a summary of quality check data including binding density, image quality, limit of detection, and more, for each sample calculated using NanoString nSolver and nCounter Analysis (T7).
(XLSX)

**S2 Table. This file contains results from statistical analyses performed and summaries within the results of this manuscript.** Inside the file contains lists all normalised gene counts that passed quality checks and thresholding with corresponding samples including base mean, $\log_2$ fold change, unadjusted p-values, adjusted p-values, false discovery rate and significance rank (T1); filtered list of only significant differentially expressed genes (T2); full results pertaining to the gene set analysis performed using Rosalind Bio (T3); standardised effect sizes (Cohen's D) for all statistically significant differentially expressed genes calculated using R package "RNASeqPower" and "effsize"; lists the effect of metadata variables Age and BMI on gene expression profiles across cohorts (GWI or HC) using multivariate analysis performed with SPSS (T5).
(XLSX)

**S3 Table. File provides immune cell type abundance scores using gene expression information.** The $\log_2$ cell type abundance scores are derived from cell-specific genes using Rosalind Bio. This data was used to generate Figure 2 of the results.
(XLSX)

**S4 Table. This file documents functional and pathway interpretation for differentially expressed genes using QIAGEN IPA according to [22].** File contains Diseases and Functions according to differential gene expression including p-values and molecules (T1); lists canonical pathways along with corresponding molecules, direction of expression and pathway activity predictions including p-values, z-score, ratio and molecules (T2); causal network analyses predicting upstream and downstream regulators and networks involving specific gene families including direction of activation or inhibition, p-value, and molecules (T3); lists predicted upstream regulators according to observed gene expression changes with corresponding p-values and molecules (T4); underlying network molecules and key diseases or functions used for Figure 3 included in the present manuscript detailing key molecules (T5).
(XLSX)

## Acknowledgments

NanoString dataset was generated by the Centre of Excellence in Spatial Biology, Griffith University. The authors acknowledge NCNED researchers Jacob Batham and Chandi Magawa for their assistance with PBMC isolation and RNA extraction. The authors acknowledge all participants who donated their time.

## Author contributions

**Conceptualization:** Natalie Eaton-Fitch, Etianne Martini Sasso, Sonya Marshall-Gradisnik.

**Data curation:** Natalie Eaton-Fitch.

**Formal analysis:** Natalie Eaton-Fitch, Sonya Marshall-Gradisnik.

**Funding acquisition:** Sonya Marshall-Gradisnik.

**Investigation:** Natalie Eaton-Fitch.

**Methodology:** Natalie Eaton-Fitch, Etianne Martini Sasso, Sonya Marshall-Gradisnik.

**Project administration:** Natalie Eaton-Fitch, Etianne Martini Sasso, Sonya Marshall-Gradisnik.

**Supervision:** Sonya Marshall-Gradisnik.

**Visualization:** Natalie Eaton-Fitch.

**Writing – original draft:** Natalie Eaton-Fitch, Etianne Martini Sasso, Sonya Marshall-Gradisnik.

**Writing – review & editing:** Natalie Eaton-Fitch, Etianne Martini Sasso, Sonya Marshall-Gradisnik.

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
