## [Decision Letter · Decision Letter 0]

16 Jul 2025

PONE-D-25-24742

Immune transcriptomic changes in Australian Gulf War veterans.

PLOS ONE

Dear Dr. Fitch,

Thank you for submitting your manuscript to PLOS ONE. After careful consideration, we feel that it has merit but does not fully meet PLOS ONE’s publication criteria as it currently stands. Therefore, we invite you to submit a revised version of the manuscript that addresses the points raised during the review process.

We look forward to receiving your revised manuscript.

Kind regards,

Seth Agyei Domfeh, PhD

Academic Editor

PLOS ONE

Journal Requirements:

“Griffith University Disability and Rehabilitation Program”

Please state what role the funders took in the study.  If the funders had no role, please state: 'The funders had no role in study design, data collection and analysis, decision to publish, or preparation of the manuscript.'

6. Please include captions for your Supporting Information files at the end of your manuscript, and update any in-text citations to match accordingly. Please see our Supporting Information guidelines for more information: http://journals.plos.org/plosone/s/supporting-information .

7. We note that Supplementary Material 4.xlsx in your submission contain copyrighted table. All PLOS content is published under the Creative Commons Attribution License (CC BY 4.0), which means that the manuscript, images, and Supporting Information files will be freely available online, and any third party is permitted to access, download, copy, distribute, and use these materials in any way, even commercially, with proper attribution. For more information, see our copyright guidelines: http://journals.plos.org/plosone/s/licenses-and-copyright.

 1. You may seek permission from the original copyright holder of Supplementary Material 4.xlsx to publish the content specifically under the CC BY 4.0 license.

 Please upload the completed Content Permission Form or other proof of granted permissions as an 'Other' file with your submission.

 2. If you are unable to obtain permission from the original copyright holder to publish these tables under the CC BY 4.0 license or if the copyright holder’s requirements are incompatible with the CC BY 4.0 license, please either i) remove the figure or ii) supply a replacement figure that complies with the CC BY 4.0 license. Please check copyright information on all replacement figures and update the figure caption with source information. If applicable, please specify in the figure caption text when a figure is similar but not identical to the original image and is therefore for illustrative purposes only.

8.If the reviewer comments include a recommendation to cite specific previously published works, please review and evaluate these publications to determine whether they are relevant and should be cited. There is no requirement to cite these works unless the editor has indicated otherwise. 

Reviewers' comments:

Reviewer's Responses to Questions

**Comments to the Author**

1. Is the manuscript technically sound, and do the data support the conclusions?

Reviewer #1: Yes

Reviewer #2: Yes

Reviewer #3: Yes

2. Has the statistical analysis been performed appropriately and rigorously? 

Reviewer #1: Yes

Reviewer #2: Yes

Reviewer #3: Yes

3. Have the authors made all data underlying the findings in their manuscript fully available?

Reviewer #1: Yes

Reviewer #2: No

Reviewer #3: Yes

4. Is the manuscript presented in an intelligible fashion and written in standard English?

Reviewer #1: Yes

Reviewer #2: Yes

Reviewer #3: Yes

5. Review Comments to the Author

Reviewer #1: the article has been modified as per the suggestions and may be considered for publication

check the statistical power and generalizability of the finding

lacking protein-level or cellular-level validation

Reviewer #2: General comments:

Overall this is a useful albeit relatively small contribution that I think is a worthwhile addition to the literature. I have a few small suggestions.

Abstract:

Consider re-wording “In this study, gene expression analysis investigated immune exhaustion in GWI, for the first time” since immune exhaustion was not actually a signal supported by the data (I realize that the panel is described as such, but that is made evidence in the Methods section of the abstract). Perhaps something like “In this study, we report for the first time expression-based analysis of a panel of 598 immune function-related genes in GWI”?

Introduction:

The section including the statement that “These toxic agents act as potent immunotoxins” should be qualified. The listed agents have many effects, not just immunological, and many of the “agents” listed are actually categories comprising hundreds or thousands of chemicals/agents (eg insecticides, smoke). Different insecticides work by many different mechanisms, as do particulate matter, PAHs, dioxins, metals and other burn pit smoke constituents, so it doesn’t make mechanistic sense to batch them. For example, the authors could either focus more specifically on insecticides to which (to the best of our limited knowledge) GW veterans were most exposed and cite relevant immune-related literature on those, or simply soften/broaden the statement somewhat.

The abstract mentions cellular stress and metabolic impacts, and these are discussed in the Results, so I would suggest a brief mention of these in the introduction as well (what we know about how these are altered in GWI, and by deployment-related exposures).

Methods:

Why were two methods used for RNA extraction? Could this have introduced any bias into the results?

Nanodrop readings indicate concentration and purity (and purity measurement ranges should be reported—ie 260/280 and 230/260 ratios), but not quality in the sense of RNA integrity.

Do the authors have any concerns about age skewing the results, given the lower average ages of the HC individuals (and apparently at least one much younger participant, at 18)?

Results:

The authors report SEM instead of SD for population demographic characteristics, I assume because they are comparing these categories statistically. However, since reporting only the SEM and not SD or range makes it difficult to evaluate the population distributions, I recommend also reporting the ranges and/or SDs.

Are the p values for each gene corrected in any way for multiple comparisons? “This panel contains 785 genes to elucidate mechanisms behind T cell, B cell and NK cell exhaustion in disease.” Is struck out in Track Changes in the manuscript, but I think should be retained—this is useful information that readers may wish to easily access, without going to Suppl File 1. But—now that I have opened that file—it has fewer than 600 genes listed. Why not 785, if this is raw data?

Figures 2 and 3 were too low resolution for me to read (Figure 1 too, but I think the take-home was obvious). I could more or less evaluate Figure 2 (the color code along the top was unclear), but Figure 3 was illegible.

Discussion:

Similar to my earlier comment about the Abstract, consider rewording “This novel study investigates altered gene expression related to immune exhaustion” to “This novel study investigates altered gene expression related to immune function”

The paragraph on mitochondrial/metabolic alterations could be expanded a bit, and discussed in relation to the immune function focus of this manuscript. The possible change in fatty acid metabolism makes me think of these publications: https://pmc.ncbi.nlm.nih.gov/articles/PMC6660083/
https://pubmed.ncbi.nlm.nih.gov/37438460/ . Altered ATAD3A levels suggest mitochondrial involvement and engagement of the cGAS STING pathway, thereby relating as well to possible inflammation (eg, https://pubmed.ncbi.nlm.nih.gov/34333111/ ). There is also a literature on mitochondrial DNA damage and copy number alteration that may be relevant, since this outcome was mentioned in the context of SESN2 downregulation.

Data availability:

I thought that PLOS ONE’s policy was that all transcriptomic data should be uploaded to GEO?

Reviewer #3: This is an interesting and valuable study. Very few labs work with human samples from GWI patients. Data analysis was done thoroughly, and conclusions are supported by data.

However, there are some issues.

1) How did authors deal with genes that had low counts? How background was determined? Was background subtracted?

2) How healthy controls (HC) were selected? Were HC sedentary? Are/were HCs in military service? As I understand from the manuscript, HCs were not matched with GWI patients by age and BMI. There were no GWI patients younger than 50 years old in 2024. And it is written in the manuscript that participants’ age was from 18 to 65. Definitely, some healthy controls were very young. Do authors consider that age differences between GWI patients and HC could be the reason for some differences in gene expression? The same goes for difference in BMI between GWI patients and HCs. Authors should discuss this issue.

3) Authors should describe more the treatment of participants at the time of blood collection. Were participants fasting? Did they rest before blood collection? This is critical for the comparison of results of different studies.

4) Authors should add the comparison of their results with the results of transcriptomic GWI studies from other labs, for example, with earlier microarrays results and with the later RNA-seq results on animal GWI model and human GWI transcriptomic data. Direct comparisons (differentially expressed genes) probably are not possible, but comparison on the level of pathways and gene ontologies will strengthen the manuscript.

6. PLOS authors have the option to publish the peer review history of their article (what does this mean? ). If published, this will include your full peer review and any attached files.

**Do you want your identity to be public for this peer review?** For information about this choice, including consent withdrawal, please see our Privacy Policy .

Reviewer #1: No

Reviewer #2: No

Reviewer #3: **Yes: ** Lubov Nathanson

---

## [Author Response · Author response to Decision Letter 1]

24 Oct 2025

26th August 2025

Response to Reviewers

Dear Editorial Team, PlosOne

Regarding submission of manuscript: PONE-D-25-24742 “Immune transcriptomic changes in Australian Gulf War veterans.”

Thank you for your prompt return of the review for the abovementioned manuscript. The authors return the revisions of the abovementioned manuscript prior to the deadline of August 29th.

A total of 16 comments across three reviewers were identified by the author team. Of the 16 comments, 13 were accepted for modifications incorporated in the manuscript. The remaining 3 comments did not suggest direct changes to the manuscript, and therefore, clarification was provided for the reviewer. All changes made are outlined in the following pages accompanied by the revised manuscript (clean and track changes documents). Additional files submitted in this submission include updated figures for resolution and updated supplementary materials.

We sincerely thank you for your review and time.

Regards,

Dr Natalie Eaton-Fitch.

National Centre for Neuroimmunology and Emerging Diseases,

Griffith University, Australia

Reviewers' comments:

Reviewer's Responses to Questions

Comments to the Author

1. Is the manuscript technically sound, and do the data support the conclusions?

Reviewer #1: Yes

Reviewer #2: Yes

Reviewer #3: Yes

2. Has the statistical analysis been performed appropriately and rigorously?

Reviewer #1: Yes

Reviewer #2: Yes

Reviewer #3: Yes

3. Have the authors made all data underlying the findings in their manuscript fully available?

Reviewer #1: Yes

Reviewer #2: No

Reviewer #3: Yes

4. Is the manuscript presented in an intelligible fashion and written in standard English?

Reviewer #1: Yes

Reviewer #2: Yes

Reviewer #3: Yes

Reviewer #1:

Comment 1:

The article has been modified as per the suggestions and may be considered for publication check the statistical power and generalizability of the finding lacking protein-level or cellular-level validation.

Author Response to Reviewer 1 Comment 1:

Comment accepted and manuscript modified.

The authors thank Reviewer 1 for their comment and as per their suggestion have checked the statistical power of the findings. Using the R package “RNASeqPower” it was determined that 7 participants per group were required to meet statistical power for samples with lower effect sizes. Effect sizes were determined using R package “effsize” using normalised expression mean intensity values of healthy controls and veterans with GWI and computed log2 fold change. The effect sizes are included below which demonstrate a moderate to large effect size for the majority of significant genes. The effect size values have now been included within the uploaded supplementary material 1 and a statement in the Discussion (lines 426-428) has been included which reads: “Nevertheless, effect sizes calculated for each differentially expressed gene were found to be moderate to large demonstrating a sufficient sample size to support these findings. With further investigations incorporating larger sample sizes and protein-level validation may also be considered for future research.”

While this current investigation did not complete protein or cellular level validation, the findings demonstrate the necessity of further investigations in this area in which additional funding and resources may be leveraged. A sentiment shared in subsequent reviewer comments.

Gene Cohen’s d

Upregulated

SIGLEC1 0.325

BPI 0.298

MMP9 0.927

RSAD2 0.256

CEACAM1 0.731

IFIT1 0.119

IFIT3 0.317

CXCL1/2/3 0.424

PTGS2 0.322

CEACAM3 1.202

ITGB3 0.934

EGF 0.566

LTBP1 0.941

GREM2 0.531

OAS3 0.486

IL1B 0.372

ELOVL7 0.843

MX1 0.427

TNFAIP3 0.643

FCAR 0.791

MX2 0.68

Downregulated

TRDV3 -0.884

IGHG1 -0.302

TRGV4 -0.219

TRDV1 -0.817

TRDV2 -0.966

IL7 -0.945

IGHV4-59 -0.446

EHHADH -0.945

IDO1 -0.731

CXCR6 -0.922

TRGC1 -0.654

SESN2 -0.739

Reviewer #2:

Comment 1:

Overall this is a useful albeit relatively small contribution that I think is a worthwhile addition to the literature. I have a few small suggestions.

Abstract: Consider re-wording “In this study, gene expression analysis investigated immune exhaustion in GWI, for the first time” since immune exhaustion was not actually a signal supported by the data (I realize that the panel is described as such, but that is made evidence in the Methods section of the abstract). Perhaps something like “In this study, we report for the first time expression-based analysis of a panel of 598 immune function-related genes in GWI”?

Author Response to Reviewer 2 Comment 1:

Comment accepted, manuscript modified.

The authors thank Reviewer 2 for their comment. While a limitation presented in the manuscript was the sample size of the cohort, the authors agree that this does not discount the contribution to the literature within this area. As an area of unmet need in an aging population, research into the pathomechanisms of Gulf War Illness remains critical to address growing public health concerns. Further, sample size calculations suggest that a sample size of seven per cohort was sufficient for statistical power to observe differential gene expression in those identified.

To address the primary suggestion of this comment, the abstract has been amended to read:

Lines 20-21 of the track changes manuscript: “In this novel study, we report gene expression-based analysis of a panel of 785 immune function related gene markers in GWI.”

Please note that the suggested “598” has been corrected to “785” to accurately demonstrate probes included in the Immune Exhaustion panel used. The discrepancy between the number of probes and the panel and the number of analysed genes is due to the exclusion of data that did not meet background thresholds.

Comment 2:

Introduction: The section including the statement that “These toxic agents act as potent immunotoxins” should be qualified. The listed agents have many effects, not just immunological, and many of the “agents” listed are actually categories comprising hundreds or thousands of chemicals/agents (eg insecticides, smoke). Different insecticides work by many different mechanisms, as do particulate matter, PAHs, dioxins, metals and other burn pit smoke constituents, so it doesn’t make mechanistic sense to batch them. For example, the authors could either focus more specifically on insecticides to which (to the best of our limited knowledge) GW veterans were most exposed and cite relevant immune-related literature on those, or simply soften/broaden the statement somewhat.

Author Response to Reviewer 2 Comment 2:

Comment accepted, manuscript modified.

The authors thank Reviewer 2 for their comment and agree that the original introduction of the manuscript was somewhat generalised to categorise chemicals and agents. The authors have briefly expanded their explanation in the introduction guided by the above comments without diverting from the required information to demonstrate rationale for this research. The introduction now reads:

Lines 57-66 of the track changes manuscript: “Current evidence supports the hypothesis that a combination of toxic chemical and environmental agents, including insecticides, smoke from oil-well fires, pyridostigmine bromide (PB) result in a veteran developing GWI [4]. These toxic environmental and chemical agents are found to be statistically associated with immune system function [5,6]. While intracellular mechanisms remain diverse, agents including organophosphates and carbamate insecticides and PB, used prophylactically by veterans, as well as products from oil well fires, including particulate matter, heavy metals and polycyclic aromatic hydrocarbons are linked with chronic inflammation, oxidative stress and neuronal damage [7–10]. Therefore, demonstrating that no single exposure results in the occurrence of GWI. ”

Included references:

[4] White RF, Steele L, O’Callaghan JP, Sullivan K, Binns JH, Golomb BA, et al. Recent research on Gulf War illness and other health problems in veterans of the 1991 Gulf War: Effects of toxicant exposures during deployment. Cortex 2016;74:449–75. https://doi.org/10.1016/j.cortex.2015.08.022.

[5] Bou Zerdan M, Moussa S, Atoui A, Assi HI. Mechanisms of Immunotoxicity: Stressors and Evaluators. Int J Mol Sci 2021;22:8242. https://doi.org/10.3390/ijms22158242.

[6] Semwal R, Semwal RB, Lehmann J, Semwal DK. Recent advances in immunotoxicity and its impact on human health: causative agents, effects and existing treatments. Int Immunopharmacol 2022;108:108859. https://doi.org/10.1016/j.intimp.2022.108859.

[7] Mitra A, Sarkar M, Chatterjee C. Modulation of Immune Response by Organophosphate Pesticides: Mammals as Potential Model. Proc Zool Soc 2019;72:13–24. https://doi.org/10.1007/s12595-017-0256-5.

[8] Burzynski HE, Macht VA, Woodruff JL, Crawford JN, Erichsen JM, Piroli GG, et al. Pyridostigmine bromide elicits progressive and chronic impairments in the cholinergic anti-inflammatory pathway in the prefrontal cortex and hippocampus of male rats. Neurobiol Stress 2022;18:100446. https://doi.org/10.1016/j.ynstr.2022.100446.

[9] Joyce MR, Holton KF. Neurotoxicity in Gulf War Illness and the potential role of glutamate. Neurotoxicology 2020;80:60–70. https://doi.org/10.1016/j.neuro.2020.06.008.

[10] Brooks AW, Sandri BJ, Nixon JP, Nurkiewicz TR, Barach P, Trembley JH, et al. Neuroinflammation and Brain Health Risks in Veterans Exposed to Burn Pit Toxins. Int J Mol Sci 2024;25:9759. https://doi.org/10.3390/ijms25189759.

Comment 3:

[Introduction:] The abstract mentions cellular stress and metabolic impacts, and these are discussed in the Results, so I would suggest a brief mention of these in the introduction as well (what we know about how these are altered in GWI, and by deployment-related exposures).

Author Response to Reviewer 2 Comment 3:

Comment accepted, manuscript modified.

The authors thank Reviewer 2 for their comment, in the drafting of the original manuscript, the introduction focused on immunological disturbances in GWI to provide rationale to conduct this investigation. However, aligning with the suggestions made, the authors have amended the introduction to incorporate background on cellular stress and metabolic impacts in relation to immune cell disturbances in veterans living with GWI. The introduction now reads:

Lines 67-80 of the track changes manuscript: “One mechanism may include the inhibition of acetylcholinesterase by insecticides and PB resulting in an accumulation of acetylcholine which consequentially influences cellular metabolism and immunological functions, including reactive oxygen species (ROS), chronic inflammation, impaired cytotoxic pathways, cytokine production and immune cell activation [7–10]. As an example, an investigation into the effect of the pesticide permethrin with PB on a GWI mice model reported increased activation of both peripheral and brain adaptive immune responses [11]. Further, the accumulation of acetylcholine is further linked with oxidative stress. In an experimental GWI mice model, impaired cellular metabolism promoting ROS have been reported in association with immune dysregulation [16]. This is further evidenced in diverse immunological studies reporting lymphocyte disturbances, altered lymphocyte subsets, interleukin (IL) and cytokine production, and production of antibodies are modified in veterans with GWI [12–15].”

Included references:

[7] Mitra A, Sarkar M, Chatterjee C. Modulation of Immune Response by Organophosphate Pesticides: Mammals as Potential Model. Proc Zool Soc 2019;72:13–24. https://doi.org/10.1007/s12595-017-0256-5.

[8] Burzynski HE, Macht VA, Woodruff JL, Crawford JN, Erichsen JM, Piroli GG, et al. Pyridostigmine bromide elicits progressive and chronic impairments in the cholinergic anti-inflammatory pathway in the prefrontal cortex and hippocampus of male rats. Neurobiol Stress 2022;18:100446. https://doi.org/10.1016/j.ynstr.2022.100446.

[9] Joyce MR, Holton KF. Neurotoxicity in Gulf War Illness and the potential role of glutamate. Neurotoxicology 2020;80:60–70. https://doi.org/10.1016/j.neuro.2020.06.008.

[10] Brooks AW, Sandri BJ, Nixon JP, Nurkiewicz TR, Barach P, Trembley JH, et al. Neuroinflammation and Brain Health Risks in Veterans Exposed to Burn Pit Toxins. Int J Mol Sci 2024;25:9759. https://doi.org/10.3390/ijms25189759.

[11] Joshi U, Pearson A, Evans JE, Langlois H, Saltiel N, Ojo J, et al. A permethrin metabolite is associated with adaptive immune responses in Gulf War Illness. Brain, Behavior, and Immunity 2019;81:545–59. https://doi.org/10.1016/j.bbi.2019.07.015.

[12] Shetty GA, Hattiangady B, Upadhya D, Bates A, Attaluri S, Shuai B, et al. Chronic Oxidative Stress, Mitochondrial Dysfunction, Nrf2 Activation and Inflammation in the Hippocampus Accompany Heightened Systemic Inflammation and Oxidative Stress in an Animal Model of Gulf War Illness. Front Mol Neurosci 2017;10. https://doi.org/10.3389/fnmol.2017.00182.

[13] Whistler T, Fletcher MA, Lonergan W, Zeng X-R, Lin J-M, LaPerriere A, et al. Impaired immune function in Gulf War Illness. BMC Medical Genomics 2009;2:12. https://doi.org/10.1186/1755-8794-2-12.

[14] Vojdani A, Thrasher JD. Cellular and humoral immune abnormalities in Gulf War veterans. Environ Health Perspect 2004;112:840–6. https://doi.org/10.1289/ehp.6881.

[15] Broderick G, Kreitz A, Fuite J, Fletcher MA, Vernon SD, Klimas N. A pilot study of immune network remodeling under challenge in Gulf War Illness. Brain Behav Immun 2011;25:302–13. https://doi.org/10.1016/j.bbi.2010.10.011.

[16] Sultana E, Shastry N, Kasarla R, Hardy J, Collado F, Aenlle K, et al. Disentangling the effects of PTSD from Gulf War Illness in male veterans via a systems-wide analysis of immune cell, cytokine, and symptom measures. Military Medical Research 2024;11:2. https://doi.org/10.1186/s40779-023-00505-4.

Comment 4:

Methods: Why were two methods used for RNA extraction? Could this have introduced any bias into the results?

Author Response to Reviewer 2 Comment 4:

Clarification provided, manuscript not modified.

The authors thank the reviewer for their comment and wish to provide clarification. A small number of samples (n=2) were extracted using the trizol method from a previous research project. Nanodrop readings indicated not only a high yield, but also a high purity. Therefore, any potential risk arising from an alternative method (being reagent contaminants) has been removed. For the reviewer’s interest, the 260/280 and 260/230 readings for the two samples are below demonstrating high purity meeting NanoString requirements:

Sample Name 260/280 260/230

HC10 1.98 1.99

HC11 1.90 1.23

Comment 5:

[Methods:] Nanodrop readings indicate concentration and purity (and purity measurement ranges should be reported—ie 260/280 and 230/260 ratios), but not quality in the sense of RNA integrity.

Author Response to Reviewer 2 Comment 5:

Comment accep

---

## [Decision Letter · Decision Letter 1]

19 Nov 2025

Immune transcriptomic changes in Australian Gulf War veterans.

PONE-D-25-24742R1

Dear Dr. Fitch,

We’re pleased to inform you that your manuscript has been judged scientifically suitable for publication and will be formally accepted for publication once it meets all outstanding technical requirements.

Kind regards,

Seth Agyei Domfeh, PhD

Academic Editor

PLOS ONE

Additional Editor Comments (optional):

Reviewers' comments:

Reviewer's Responses to Questions

**Comments to the Author**

1. If the authors have adequately addressed your comments raised in a previous round of review and you feel that this manuscript is now acceptable for publication, you may indicate that here to bypass the “Comments to the Author” section, enter your conflict of interest statement in the “Confidential to Editor” section, and submit your "Accept" recommendation.

Reviewer #1: All comments have been addressed

Reviewer #2: All comments have been addressed

Reviewer #3: All comments have been addressed

2. Is the manuscript technically sound, and do the data support the conclusions?

Reviewer #1: Yes

Reviewer #2: Yes

Reviewer #3: Yes

3. Has the statistical analysis been performed appropriately and rigorously? 

Reviewer #1: Yes

Reviewer #2: Yes

Reviewer #3: Yes

4. Have the authors made all data underlying the findings in their manuscript fully available?

Reviewer #1: Yes

Reviewer #2: Yes

Reviewer #3: Yes

5. Is the manuscript presented in an intelligible fashion and written in standard English?

Reviewer #1: Yes

Reviewer #2: Yes

Reviewer #3: Yes

6. Review Comments to the Author

Reviewer #1: Manuscript Title: Immune transcriptomic changes in Australian Gulf War veterans.

the article has been modified as per teh suggestions and may be considered for publication

Reviewer #2: (No Response)

Reviewer #3: (No Response)

7. PLOS authors have the option to publish the peer review history of their article (what does this mean? ). If published, this will include your full peer review and any attached files.

**Do you want your identity to be public for this peer review?** For information about this choice, including consent withdrawal, please see our Privacy Policy .

Reviewer #1: No

Reviewer #2: **Yes: ** Joel Meyer

Reviewer #3: **Yes: ** Lubov Nathanson

---

## [Editor Report · Acceptance letter]

PONE-D-25-24742R1

PLOS ONE

Dear Dr. Fitch,

I'm pleased to inform you that your manuscript has been deemed suitable for publication in PLOS ONE. Congratulations! Your manuscript is now being handed over to our production team.

Kind regards,

on behalf of

Dr. Seth Agyei Domfeh

Academic Editor

PLOS ONE